# Phytochemical Analysis, Biochemical and Mineral Composition and GC-MS Profiling of Methanolic Extract of Chinese Arrowhead *Sagittaria trifolia* L. from Northeast China

**DOI:** 10.3390/molecules24173025

**Published:** 2019-08-21

**Authors:** Maqsood Ahmed, Mingshan Ji, Aatika Sikandar, Aafia Iram, Peiwen Qin, He Zhu, Ansar Javeed, Jamil Shafi, Zeeshan Iqbal, Mazher Farid Iqbal, Zhonghua Sun

**Affiliations:** 1College of Plant Protection, Shenyang Agricultural University, Shenyang 110866, China; 2Government of the Punjab Education Department, Gujranwala 52250, Pakistan; 3College of Biosciences and Biotechnology, Shenyang Agricultural University, Shenyang 110866, China; 4Department of Plant Pathology, University of Agriculture Faisalabad, Faisalabad 38040, Pakistan; 5Department of Animal Sciences, University of Sargodha, Sargodha 40100, Pakistan

**Keywords:** *Sagittaria trifolia*, phytochemicals, phenol and flavonoids content, DPPH, biochemical composition

## Abstract

*Sagittaria trifolia* is a medicinal foodstuff of China and East Asia belonging to the family Alismataceae. Samples of *S. trifolia* tubers were collected from Meihekow, Siping, Jilin, Harbin and Wuchang from Northeast China. The current study was aimed to evaluate the qualitative and quantitative analysis, antioxidant activity, biochemical analysis and chemical composition of different populations of *S. trifolia.* By using Folin–Ciocalteu, aluminium chloride colourimetric and 1,1-diphenyl-1-picrylhydrazyl (DPPH), total phenol and flavonoids content and antioxidant activity was analysed. Furthermore, chemical composition, biochemical analysis and mineral substances were also determined. The results showed the presence of flavonoids, phenols, saponins, tannins, glycosides and steroids except for alkaloids and terpenoids by qualitative analysis. Quantitative analysis revealed that highest total phenol, flavonoids content and antioxidant potential identified from Meihekow, i.e., 2.307 mg GAE/g, 12.263 mg QE/g and 77.373%, respectively. Gas chromatography-mass spectrometry results showed the presence of 40 chemical compounds corresponding to 99.44% of total extract that might be responsible for antioxidant properties. Mineral and biochemical analysis revealed the presence of calcium, magnesium, potassium, sodium, iron, copper, zinc and, carbohydrate, protein, fibre and fat contents, respectively. Interestingly, all *S. trifolia* populations collected from different locations possess similar composition. The dietary values, phytoconstituents, antioxidant activities and nutritional and curative chemical compounds of *S. trifolia* are beneficial for the nutritherapy of human beings.

## 1. Introduction

Free radicals and other reactive oxygen species are those agents which are implicated in many acute and chronic diseases, like diabetes, asthma, Parkinson’s, atherosclerosis, cancer, cataracts, neurodegenerative disorder, liver injury as well as humans aging [1,2]. However, antioxidants are beneficial substances that are involved in delaying and inhibition of such agents by avoiding oxidative damage to the target sites [3]. Antioxidants have the ability to trap free radicals, such as peroxide and hydroperoxide, thereby initiating the mechanism of oxidation leading towards degenerative disorders [4]. Moreover, phytochemicals extracted from natural plants are considered to be safe and good alternatives when compared to synthetic antioxidants. They contain antioxidants in the form of flavonoids, tannins, phenols and proanthocyanidins which are involved in the reduced mortality caused by degenerative disarray [5].

Natural phytochemicals contained in plants, such as alkaloids, flavonoids, phenols and tannins, are regarded as the molecules with the ability to neutralize free radicals and have gained increased attention of researchers and consumers as potential antioxidants. Previous data also showed phenolic contents from plant resources show anti-mutagenic, anti-carcinogenic and antioxidant activity [6]. Aquatic edible plants also contribute towards an extra source of food and vegetables possessing high curative properties. Moreover, vegetables are the edible portions of herbaceous plants that are consumed as raw (main dish or as salad) and by cooking, which may be sweet, bitter, or tasteless [7]. Vegetables and fruits are vital sources of protecting the body from various diseases by elevating the normal functioning of the body system, regulating metabolic activities, tissue repair and weight maintenance which is linked to reduced risk of chronic diseases [8,9,10].

*Sagittaria trifolia* is widely distributed in most of rice growing areas in the world. It is also called Chinese arrowhead, arrowroot, three-leaf plant species, is a native herb in China and is also found across the sopping parts of Europe and Asia native to Northern Taiwan [11]. *S. trifolia* have a wetland habitat and is an important herb in rice crops. However, ecological distribution of *S. trifolia* is widespread in Taiwan, Macao and Hong Kong. It is also found in Afghanistan, India, Cambodia, Indonesia, Malaysia, Japan, Thailand, Kazakhstan, Turkmenistan, Kyrgyzstan, Russia, Myanmar, Nepal, Iran, Philippines, Uzbekistan and Vietnam [12].

It is a well-known herb species, a nutritional foodstuff and is innocuous in appearance and use. Lii and Chang [13] reported the usage of this herb species as an alternative to unprocessed raw material of mung bean starch and for the preparation of noodles, different processed food preparations and its corm contained 10–20% starch of total fresh weight [14,15]. *Sagittarius* products are processed and exported from China to different Asian countries as an off-season vegetable [16]. Cooked tubers and petioles are used for edible purposes in Manipur and Southeast Asia while, in Vietnam, these parts of *S. trifolia* are used in the preparation of soup [17]. In China, tubers of *S. trifolia* have been used in traditional Chinese herbal medicines and as vegetables since ancient times. Previous reports highlighted the ability of *S. trifolia* to absorb phosphorus and nitrogen from eutrophic water explaining its importance in water purification [18]. The proximate composition of dried root/tubers revealed that they consist of 364 calories/100g, i.e., carbohydrate 76.2 g, protein 17 g, fibre 3.1 g, ash 5.8 g, fat 1 g, phosphorus 561 mg, calcium 44 mg, potassium 2480 mg, iron 8.8 mg, thiamine 0.54 mg, riboflavin 0.14 mg, niacin 4.76 mg and ascorbic acid 17 mg, with no carotene [19]. The composition of *S. trifolia* corms per 100 g basis was reported by Leung et al. [20] as energy 107 cal, fat 0.3 g, protein 5 g, fibre 0.9 g, carbohydrate 22.4 g, ash 1.7 g, with a mineral composition as P 165 mg, Ca 13 mg, K 729 mg, Fe 2.6 mg, riboflavin 0.04 mg, thiamine 0.16 mg, niacin 1.4 mg, moisture 70.6% and ascorbic acid 5 mg. Leaves of this herb are used to cure several skin diseases whereas, corms are used as galactofuge, discutient and may induce premature birth in human beings [19]. Essential oil extracted from the above-ground portion of *S. trifolia* has been used in herbal medicines for the curing of wounds, headache, digestive disorders and rheumatism [21]. Similarly, in Vietnam it is used for the treatment of dizziness or pimples, while aerial parts are used as fodder in various regions of Southeast Asia and India [17]. Sugimoto et al. [22] determined amylose from three cultivars of *S. trifolia* were in the range from 28.2–29.9% and 26.6–31.3% by amperometric iodotitrimetry and gel permeation chromatography (GMC), respectively. Moreover, Li [23] reported vitamin B, asparagines, glucose, d-fructose and d-glactose from this plant. It contains diterpenoid with sagittine H, sclareol and ent-kaur-16-ene-19-oate reported by Liu et al. [24]. Antimicrobial activity from *S. trifolia* was reported from this plant, showing its medicinal importance in herbal medicine use during childbirth and remediation for skin diseases [21]. Although some studies have been reported on the biochemical composition of corms of *S. trifolia* with therapeutic potential, while extensive study on phytochemical analysis and proximate compositions of vegetative/aerial parts has not been studied to date. Therefore, the current study was aimed to determine the:presence of phytoconstituents,quantification of phenol and flavonoids content,antioxidants activity, biochemical and proximate analysis,chemical composition of crude extract of leaves of *S. trifolia* populations.

## 2. Results

### 2.1. Extract Yield (%)

Extract yield was obtained from different populations of *S. trifolia* leaves by the solvent extraction technique. Methanol was employed as the solvent in the extraction process. In comparison (*p* < 0.05) the highest extract yield was produced by WC-XC, followed by MHK, GL-G, JL-O and HNW, respectively (Figure 1). The obtained extract was sticky, oily and flowable in consistency with a greenish-black colour in appearance. 

### 2.2. Qualitative Analysis

Initial phytochemical screening tests were employed for the identification of bioactive components, which ultimately lead towards drugs discovery and development. Screening tests were performed for the presence of phytoconstituents like flavonoids, alkaloids, glycosides, terpenoids, steroids, flavones, tannins, phenols and saponins from leaves of *S. trifolia* populations. Screening tests for phytochemicals confirmed the existence of tannins, steroids, flavonoids and flavones, phenols and saponins, whereas alkaloids and terpenoids were not found during the analysis (Table 1).

### 2.3. Quantitative Analysis

#### 2.3.1. Total Phenolic Content (TPC) and Total Flavonoids Content (TFC)

The total phenolic and flavonoids content extracted from leaves of *S. trifolia*, along with the DPPH scavenging potential are indicated in (Table 2). Results revealed that the maximum total phenol and flavonoids content were extracted from MHK followed by WC-XC, GL-G, HNW and JL-O, respectively. The results also showed relatively comparable phenol and flavonoids content among the populations of *S. trifolia* under investigation.

#### 2.3.2. DPPH Radical Scavenging Activity

To determine inhibition (%), 1,1-diphenyl-1-picrylhydrazyl (DPPH) was used due to the stable and free radical nature which can be dissolved easily in methanol and identified via colour absorption by spectrophotometer at 517 nm. The antioxidant molecules trap free radicals by their contribution of hydrogen molecules which changed the colour of the DPPH assay solution to light yellow and, consequently, reduced the absorbance. Data attained by DPPH inhibition (%) by scavenging activity of free radical is shown in (Table 2). Results showed that MHK demonstrated high inhibition, followed by GL-G, HNW, WC-XC and JL-O, respectively. Results were similar with respect to the inhibition percent from all populations of *S. trifolia*.

### 2.4. Gas Chromatography-Mass Spectrometry (GC-MS) Analysis

The presence of chemical components in the methanol extract of *S. trifolia* was analysed by GC-MS. Prime active components, peak area (%), molecular formula (M.F) and molecular weight (M.W) were determined (Table 3). Forty chemical compounds corresponding to 99.44% of total extract by GC fraction from *S. trifolia* leaves were identified. Moreover, phenylacetic acid, 2-methylcyclohex-2-enyl ester (34.30%), 3-octadecyne (7.72%), pentanoic acid, 2-[(phenylmethoxy) imino]-, trimethylsilyl ester (7.08%), manoyl oxide (5.62%) and 16-kaurene (5.36%) were the chief chemical compounds while 35 others compounds were present in minor quantities with peak areas ranging from 0.22–3.42%.

### 2.5. Biochemical and Mineral Composition

The results obtained from five different populations of *S. trifolia* on biochemical analysis indicated its richness in mineral element (M.E), crude fibre (C.F), carbohydrate (CHO), crude protein (C.P), dry matter (D.M) and fats (F) contents (Figure 2). Significant (*p* < 0.05) moisture content was produced by JL-O 17.32%, followed by MHK, WC-XC, GL-G and HNW as 16.61, 15.48, 14.59 and 11.60%, respectively. Results of fat content showed a significant (*p* < 0.05) effect by GL-G 0.100%, WC-XC 0.075% and JL-O 0.038%. However, JL-O showed insignificant results to HNW and MHK, with 0.032 and 0.027%, producing low fat content. Statistically significant (*p* < 0.05) protein was determined by GL-G 26.25% when compared to JL-O 23.85% and MHK 21.87%. HNW showed (*p* > 0.05) 25.37% protein value, which is insignificant to WC-XC being 25.00% proteins. Results on fibre content showed that MHK recorded statistically insignificant (*p* > 0.05) 16.50% fibre with HNW 16.65%, GL-G 16.50%, WC-XC 16.05% and JL-O 15.90%, respectively. Maximum carbohydrates were recorded by WC-XC 28.39% which were insignificant (*p* > 0.05) to JL-O 27.76% while, significant (*p* < 0.05) to all other populations. Results from biochemical analysis also showed maximum mineral matter/ash recorded by MHK, HNW and GL-G 20% showing a statistically insignificant (*p* > 0.05) relation with each other while they showed a significant affect (*p* < 0.05) in comparison to JL-O and WC-XC, having 15% ash, respectively. The results showed that all populations of *S. trifolia* are almost alike in biochemical content and minerals.

### 2.6. Mineral Composition (%)

The mineral composition was also determined, as indicated in Figure 3, which revealed that all the populations of *S. trifolia* contained considerable amounts of minerals. However, maximum calcium (Ca) was recorded from MHK followed by HNW, WC-XC and GL-G, respectively, while the minimum mineral elements were recorded by JL-O. Maximum magnesium (Mg) was observed in JL-O followed by MHK and WC-XC and GL-G, respectively. The lowest magnesium content was reported to be 0.79% from HNW. Moreover, the potassium content (K) is slightly greater with highest percentage in GL-G followed by JL-O, MHK, HNW and WC-XC, respectively. Maximum sodium (Na) was recorded by WC-XC followed by GL-G and HNW. However, minimum (Na) was produced by MHK and JL-O, respectively.

Additionally, data presented in Figure 4 reveals that the maximum iron (Fe) was recorded by WC-XC followed by GL-G, HNW, JL-O and MHK, respectively. Similarly, MHK afforded maximum copper (Cu) followed by WC-XC and GL-G, respectively. However, maximum zinc (Zn) was recorded by WC-XC, followed by GL-G, HNW and JL-O, respectively.

## 3. Discussions

Antioxidants are enormously key substances which provide protection from damage exerted by free radicals and reactive oxygen species that are associated with induced oxidative stress. Hence, the antioxidant activity, total phenol and flavonoids content and biochemical composition of the methanolic extract of *S. trifolia* populations were determined in search of new bioactive components from natural plant resources. However, extraction of secondary metabolites from natural plant resources is linked with specific plant parts and the utility of solvents for extraction. High polarity solvents are preferred as compared to low polarity solvent with increased extract yield. In our results methanol demonstrated high extract yield from all populations. Our findings are in accordance with the previous data by Rao et al. [25] that methanol-derived maximum extract yield from *S*. *sagittifolia* was 49.7 mg/20g of dried sample as compared to other solvents. Moreover, Awang et al. [26] also reported methanol and ethanol as a solvent of choice due to greater yielding ability than other solvents from leaves of *M.malabathrium.* Methanol afforded maximum extract yield from *S. sagittifolia* leaves and tubers among all other solvents [27]. Further, Ahmed et al. [28] reported higher extract yield from *C. colocynthis* and *C. sativa* leaves using methanol as an extraction solvent.

Phytochemicals or secondary metabolites are occurring naturally in plants thereby exhibit protective, curative and defensive potential. The regular intake of these bio functional constituents through dietary sources may encourage a healthy life by providing defence against various cardiovascular and chronic diseases [29]. Qualitative analysis form *S. trifolia* populations for the assessment of secondary metabolites revealed the presence of glycosides, flavonoids, flavones, tannins, saponins and steroids, except for terpenoids and alkaloids, from all populations. Additionally, all populations of *S. trifolia* present similar results for phytochemicals existence. Our results on phytochemical screening were supported by Rao and Pandey [27] that maximum phytoconstituents like alkaloids, glycosides, tannins, steroids, saponins, flavonoids, phenols, terpenoids and carbohydrates were reported from *S. sagittifolia* by using methanol, ethanol, acetone and distilled water extract.

Results from quantitative analysis for phenol and flavonoids revealed the elevated level of phenol and flavonoids form the methanol extract of *S. trifolia*. Those plants which showed richness in phenolic content are being frequently used in the foodstuff industry because of their ability to hinder with lipids oxidative degradation. Hence, they play pivotal roles in improving nutritional value and food quality [30]. Similarly, flavonoids are known to scavenge most of the oxidizing molecules and other free radicals that are considered to be the causative agents of numerous metabolic and degenerative diseases. They also trap and scavenge reactive species and protect antioxidant defence by up regulating processes [31]. However, our results on phenol and flavonoids content, 2.31 mg and 12.26 mg from *S. trifolia*, are supported by Rao et al. [25] who reported maximum phenolic and flavonoids content of 36.4 mg GAG/g and 16.60 mg QE/g from the extract of *S. sagittifolia.*

1,1-Diphenyl-1-picrylhydrazyl (DPPH) is a stable free radical which showed maximum absorbance at 517 nm and can readily go through the scavenging process by an antioxidant. However; the extent of discoloration of the solution demonstrates the scavenging activity of the antioxidant extract. All populations of *S. trifolia* showed an increased degree of inhibition percentage ranging from 63.670% to 77.347%, which were in accordance with Lilan et al. [32] who reported the content of polysaccharides to be 29.32% from *S. sagittifolia* under an optimized extraction process. It also showed good antioxidant activities against the DPPH free radical and contains reducing power. Rao et al. [25] reported the IC_50_ ranging from 18.86±0.23 to 86.65±0.43% from methanolic extract of *S. sagittifolia*, which are in agreement with our findings on *S. trifolia*. Gas chromatography-mass spectrometry analysis from methanolic extract of *S. trifolia* revealed the existence of 40 compounds. These chemical compounds are responsible for antioxidant activities. Similar, finding form chemical analysis of *S. trifolia* tubers were proposed by Yoshikawa et al. [33] who reported 14 compounds among which 11 were diterpenes, two were diterpene glucosides and one was nitroethylphenol glycoside. Moreover, 39 compounds were extracted from leaves of *S. trifolia* of which 28 compounds were identified, whereas major chemical components were hexahydrofarnesyl acetone, tetramethylhexdecenone, myristaldehyde, *n*-pentadecane and 2-hexyldecanol 62.3, 5.8, 4.7, 2.90 and 2.91%, respectively [27]. Additionally, Liu et al. [34] isolated seven new chemical compounds, e.g., *ent*-rosane diterpenoids, sagittines A-G (1–7), and one new labdane diterpene, 13-epi-manoyl oxide-19-o-α-L-2′-5′-diacetoxyarabinofuranoside from *S. sagittifolia* supported our findings on the identification of 40 chemical constituents from *S. trifolia*.

Most of the compounds contained in *S. trifolia* are important from pharmacological and botanical viewpoints, such as octadecene a hydrocarbons with long chains, alongside oleic acid, which occurs naturally in different animals and in oils and fats from vegetable sources. It is accepted as a common fatty acid in nature as monounsaturatedomega-9 [35,36]. In terms of enzymology, 16-kaurene is an enzyme like ent-kaurene synthase belongs to family lyases involved in reaction catalyzation. It also imparts in biosynthesis of diterpenoid [37]. Another paramount compound, manoyl oxide, is a member of triterpenoid compounds that are commonly used as anticancer agents [38]. Similarly, sclarene belongs to diterpene, which was reported previously in the *Podocarpus hallii* foliage and some other plants like yew trees of the genus *Taxus*. Interestingly, our results from GC-MS analysis showed that *S. trifolia* also contain such constituents which is an important pharmacological agents widely used in chemotherapy [39]. The identified compound dodecane is a fluid alkanehydrocarbon, an oily liquid, consisting of 355 isomers and used as a solvent and scintillator constituent [40]. Methyl palmitate occurs naturally in crude palm oil as a fatty acid methyl ester and medicinally, it regulates the vascular tone; however, its chemical character remains mysterious. It showed anti-fibrotic and anti-inflammatory results. Diacetyl and acetoin have a buttery taste and because of this property, manufacturers used it typically in oil-based products to make products butter-flavoured, and it is also used in the beverage industry for flavouring of wine [41,42]. Furfural is an organic compound and on its hydrogenation yielded tetrahydrofurfuryl alcohol, used in a variety of agricultural formulations, such as adjuvant in herbicides for penetration into plant parts and as insecticides [43]. However, a compound relative to furfural, i.e. 4-methoxybenzhydrazide, occurred with the same peak and area percent with furfural, commonly used in the preparation of different products and also as raw materials. It also shows biological activities, such as anticancer, antimicrobial, antiviral, anticonvulscent, antioxidant, anti-mycobacterium, antitumor, anti-inflammatory, antimalarial and herbicidal properties [44]. Among compounds possessed by *S. trifolia*, methyl pentanoate is generally used in laundry detergents, soap, beauty care and fragrances because of its fruity odour. Meanwhile, vegetables are the edible plants that are consumed in raw form and also by cooking which contain phytochemicals [7]. Similarly, aquatic edible plants are an extra source of food. However, the optimum range of moisture content for fresh vegetables is 72.4% as described by the United States Department of Agriculture (USDA) [45]. Although the moisture content reported from *S. trifolia* contains lower values as compared to USDA recommendations, whereas, the results are in line with Rao and Pandey [27], who reported 9.07±0.10% moisture content in *S.*
*sagittifolia.*

Fat content recorded from *S. trifolia* was in the range of 0.027%–0.100% from different populations are in accordance with Chang [11] who reported 0.47% crude fat in the corms of *S. trifolia*. The fat content percent of all populations of this herb are below the standard (2 g/300 g) described by the World Health Organization for vegetables [46]. The consumption of *S. trifolia* as a vegetable is beneficial for healthy and normal body functioning since use of excessive fat is associated with cardiovascular disorders. It is worth noting that those plants which contain 12% calorific value from proteins could be considered as good sources of proteins [47]. Protein values investigated from *S. trifolia* ranged from 21.876 ± 1.02–26.250 ± 1.36 are excellent sources of proteins that play a prominent role being cheap and easily available for humans. It can contribute to the daily requirement of protein (71 g/day) of lactating and pregnant mothers [48]. There were some contradictions to Chang [11] who reported 16.47% protein in corm of *S. trifolia* but, meanwhile, similar results for protein from *C. patendra*, *V. unguiculata* and *A. digitata* 15.20, 21.96, and 18.08%, respectively, were demonstrated by previous study [49]. The crude fibre content of *S. trifolia* populations 15.90 ± 0.14–16.65 ± 0.20% were above from recommended criteria 8 g/300 g by World Health Organization [46] for vegetables. High percentages of crude fibre from our results are in accordance with previous study for leafy vegetables [50]. These findings are in line with Rao et al. [25] who reported 12.08 and 12.04% crude fibre in Bitter Leaf and *Hibiscus sabdariffa,* respectively, in contradiction to Chang [11] who reported 1.76% fibre in the corms of *S. trifolia*. In addition, *S. trifolia*, being rich in fibre, would be advantageous in performing an active role in intestinal transit regulation and increasing dietary values by absorbing water, like other vegetables [51].

Complex carbohydrates found in vegetables are important, although they slowdown the conversion into simple sugars and ultimately decreasing insulin level and help in fats burning [52]. However, carbohydrates were found in the range of 22.560 ± 0.50–28.395 ± 0.25% from *S. trifolia,* which were similar to *Senna obtusfolia, Amaranthus incurvatus* and *Momordica balsamina* leaves 20, 23.7 and 39.05%, respectively [53]. Although our findings are in agreement with Patricia et al. [49] who reported 26.30% carbohydrates in *C. patendra*, but contradictory to Ooi et al. [54], who reported 46.58% carbohydrates in *Peperomia pellucida* L., which is the highest calorie contributor. Ash content from *S. trifolia* was relatively high and ranging from 15.0 ± 0.11–20.0 ± 1.33% than *A. hybridus* 8.59 ± 1.34% but less than *C. patendra* 25.67% are in accordance with recent published data [49]. These results are contradictory to Chang [11] who reported 4.76% ash in the corms of *S. trifolia*. All the populations contain high ash content within USDA standards of 0.2%–1.9%, as reported [55]. The high ash values are an indication of high mineral content. Numerous elements are required in trace amounts which are essential for a wide range of functions in the body. In view to the suggested dietary grant for calcium (1000 mg/day), magnesium (400 mg/day), phosphorus (800 mg/day) and iron (8 mg/day), *S. trifolia* could wrap dietary requirements for improving human daily diet [48]. Similarly, iron plays a vital role in oxygen binding in haemoglobin and is an imperative catalytic axis in numerous enzymes, like cytochrome oxidase. Iron also plays a role in the electron transportation within cells [56]. Moreover, hazardous effects of *S. trifolia* are not known.

In summary, this medicinal foodstuff has a wetland habitat which is cultivated in China and East Asia because of its highly nutritious tubers (protein and starch). Edibility and medicinal rating of this plant is fifth and second, respectively. Additionally, leaves of this plant are utilized as vegetables, soup formation and formulation of other processed food items. It contained a variety of chemical compounds which are important from pharmacological and curative points of view. Moreover, chemical compounds contained by this plant possess pronounced antioxidant activity, thereby used as alternative to synthetic antioxidants. In the current scenario of the fast-growing population of the world, it is necessary to find alternative sources of food and feed to fulfil the increasing demands of hygienic food for human beings globally. It is also an important plant medicinally, for nutritherapy and as a vegetable source, as well as its utilization in the formulations of other products. Finally, as natural plants they contain numerous chemical compounds which are safer and useful from edibility, medicinally and curative points of view and pose healthy impacts on human’s life worldwide. In contrast, non-green resources or synthetic preparations are harmful in various ways. Thus, natural plant populations might be taken into consideration for edible purposes and healthy life and, hence, *S. trifolia* could be a beneficial alternative. Thus, keeping in view the results of the present study, *S. trifolia* could be suggested in the daily diet to achieve nutritional requirements worldwide. Although studies of starch from corms and the essential oil composition from leaves had been carried out previously, few studies have explored the qualitative and quantitative phytochemical analysis, antioxidant activity and the biochemical composition of leaves/vegetative portions of this herb in such a comprehensive way.

## 4. Materials and Methods

### 4.1. Samples Collection

The study was conducted on five populations of *S. trifolia* collected from three provinces (Liaoning, Jilin, Heilongjiang) with five different locations of Northeast China during the year 2018. Populations with coordinates were Meihekou ‘MHK’ (42.5393° N, 125.7109° E), Siping ‘GL-G’ (43.1664° N, 124.3504° E), Jilin ‘JL-O’ (43.1520° N, 126.4371° E), Harbin ‘HNW’ (45.8038° N, 126.5350° E) and Wuchang ‘WC-XC’ (44.9143° N, 127.1500° E). The collected corms were washed using tap water to eliminate impurities and placed in refrigerator at 4 °C.

### 4.2. Herb Cultivation

The herb cultivation was performed in Greenhouse of Shenyang Agricultural University with coordinate (41.8057° N, 123.4315° E). The corms were removed from refrigerator and placed in pot with water (5–8 cm deep) for 4–5 days to initiate sprouting (Figure 5a). The well-sprouted corms were planted in separate pots (20 cm diameter) containing clay loam soil with water depth of 20 cm. Although environmental conditions did not affect the chemical and biochemical composition of the plants, however, after one week of planting, pots were removed from the greenhouse and placed in an open space for growth in a natural environment. The experiment was conducted utilizing a randomized complete block design (RCBD) using five populations (treatment) with five replications.

### 4.3. Preparation of Material

At 80 days after planting (6–7 leaves stage; Figure 5b), leaves were collected randomly from each population. The harvested leaves were washed and cleaned in tap water and placed under shade for complete drying up to 20 days. Dried leaves were grinded using mortar and pestle and extracted with methanol (4 mL g^−1^ of sample) at room temperature for 72 h. Extracts were filtered and concentrated on a rotary evaporator (Rotavapour, R-210 BUCHI Labortechnik AG, CH-9230 Flawil 1/Switzerland) to remove the solvent from the extract. Finally, extract yield was calculated using Equation (1) and the extract was stored at 4 °C in airtight glass bottles for further use.
(1)$$Extract\;yield\;=\frac{Weight\;of\;extract\;\left(g\right)}{Weight\;of\;the\;dried\;sample\;}\times 100$$

### 4.4. Phytochemical Analysis

#### Qualitative Analysis

Qualitative tests were conducted according to the standards methods to determine the presence of alkaloids, saponins, terpenoids and steroids [57], glycosides, flavonoids and flavones [58,59], phenols [60] and tannins [61].

### 4.5. Quantitative Analysis

#### 4.5.1. Total Phenolic Content (TPC)

The existence of total phenol in the crude extract of *S. trifolia* populations was assessed by Folin–Ciocalteu reagent test with some modifications. In brief, 1 mL extract (1 mg mL^−1^) was added to 2.5 mL of Folin–Ciocalteu (10%) with further the addition of 2 mL of sodium carbonate 2% (Na_2_CO_3_). The resulting mixture was incubated for 15 min in darkness and the absorbance was calculated in an ELISA 96-well plate at 765 nm by an absorbance microplate reader (SpectraMax 190, manufactured by Meigu molecular International Co. Ltd., Shanghai China; designed in California, USA). Gallic acid (1 mg mL^−1^) at concentrations of 1, 0.50, 0.25, 0.10, 0.05, 0.02, 0.01 and 0 mg/mL was used to build a standard curve to express the results as gallic acid equivalent (GAE) mgg^−1^ of extract [28]. Ten replications were analysed for each treatment.

#### 4.5.2. Total Flavonoids Content (TFC)

Total flavonoids content were examined by aluminium chloride colourimetric method. Briefly, 1 mL *S. trifolia* leaves extract was added into 3 mL methanol, 0.2 mL of 1M potassium acetate (CH_3_COOK), 0.2 mL of 10% aluminium chloride (AlCl_3_) and, finally, 5.6 mL of distilled water was added to the mixture. The resulting mixture was incubated for 30 min in darkness and absorbance was measured at 420 nm. To construct the standard curve, quercetin was used at concentrations of 1, 0.50, 0.25, 0.10, 0.05, 0.02, 0.01 and 0 mg mL^−1^. The results are represented as quercetin equivalent (QE) mg/g of total extract. Ten replications were used for each treatment.

#### 4.5.3. DPPH Radical Scavenging Activity

To measure the antioxidant activity of the crude extract of *S. trifolia*, stable radicals 1,1-diphenyl-1-picrylhydrazyl (DPPH) was used with some minor modifications. Briefly, into freshly prepared 3.5 mL DPPH solution (0.004 g/100 mL methanol), 0.5 mL of extract solution was added and incubated at room temperature in the dark for half an hour [62]. Absorbance was calculated at 517 nm and percent inhibition of DPPH solution was intended by decreasing of absorbance by following Equation (2). A lesser absorbance rate demonstrates higher radical scavenging activity:(2)$$Inhibition\;\left(\%\right)=\frac{A_{blank}-A_{sample}}{A_{blank}}\times 100$$
where: *A_blank_* = (absorbance of control); *A_sample_* = (absorbance of samples).

### 4.6. Gas Chromatography-Mass Spectrophotometry (GC-MS) Analysis

GC-MS study was conducted on an Agilent 6890-5973N USA (Agilent Technologies; Hewlett-Packard Company 2850 Centerville Road Wilmington, DE 19808-1610 USA), with the gas chromatograph set with an HP1 capillary column TG-5MS polydimethylsiloxane (30 m length × 250 µm diameter × 0.25 µm film thickness) interfaced with Hewlett Packard (5973N) mass selective detector. Parameters were; initial temperature was 70 °C (0 min) and last temperature increased to 200 °C with final time 10 °C min^−1^ while, inlet temperature was 250 °C and split ratio was 10:1. MS quadruple and thermal aux temperatures were 150 and 285 °C, respectively. The MS scan range was 35–520 units and helium was used as the carrier gas with a flow rate of 1.0 mL min^−1^. Compounds were identified and verified by comparing with gas chromatography mass spectrum literature or data provided by the National Institute of Standards and Technology Mass Spectral database Wiley/NIST.1998.1 [63]. To calculate the comparative yield of compounds raw data was followed based on gas chromatography (GC) areas with a FID correction factor.

### 4.7. Biochemical Composition and Mineral Analysis

Biochemical analysis (dry matter/moisture, fat, protein, crude fibre, carbohydrates and mineral element/ash) of *S. trifolia* populations was carried out by standard methods and are presented in Table 4.

### 4.8. Mineral Element Composition

Estimation of mineral substances in dried grinded leaves was performed by using a NOVA 400 atomic absorption spectrometer (Analytik Jena AG, Jena, Germany) hollow cathode lamps and acetylene/air flame to measure absorbance. By using slits, wavelengths and lamp current; sodium (Na), potassium (K), magnesium (Mg), zinc (Zn), calcium (Ca), copper (Cu) and iron (Fe) were calculated. The analysed results for Na, Mg, Ca and K were expressed in (%) while, Zn, Cu and Fe contents were expressed in (mg 100 g^−1^) [66]. 

### 4.9. Statistical Analysis

All the analysis for recorded data was performed using SPSS statistical software (version 18.0; Inc., Chicago, IL, USA). The analysis of variance (ANOVA) test was applied. The mean differences between treatments were calculated for significance at the *p* > 0.05 level according to Tukey’s HSD. Graphical representation of data was performed by using Sigma Plot 10.0 software.

## 5. Conclusions

Results from the present study suggested that *S. trifolia* contains a high proportion of phytoconstituents, phenol and flavonoid content thereby conferring to the free radicals’ scavenging activities and can be a potent source of natural antioxidants. Chemical composition revealed the presence of 40 compounds responsible for antioxidant potential of *S. trifolia*. Furthermore, its leaves are considered to be a good source of carbohydrates, protein, fibre, fat content and minerals, typically calcium, magnesium, potassium, sodium, iron, copper and zinc, in appreciable amount. Hence, *S. trifolia* could be a contributor towards meeting nutritional requirements of humans if used in adequate amounts and could be considered as a positive source of defence for protection against various diseases. However, further studies are desired to explore its dietary potential for the sake of commercial purpose and toxicity studies.

## Figures and Tables

**Figure 1 molecules-24-03025-f001:**
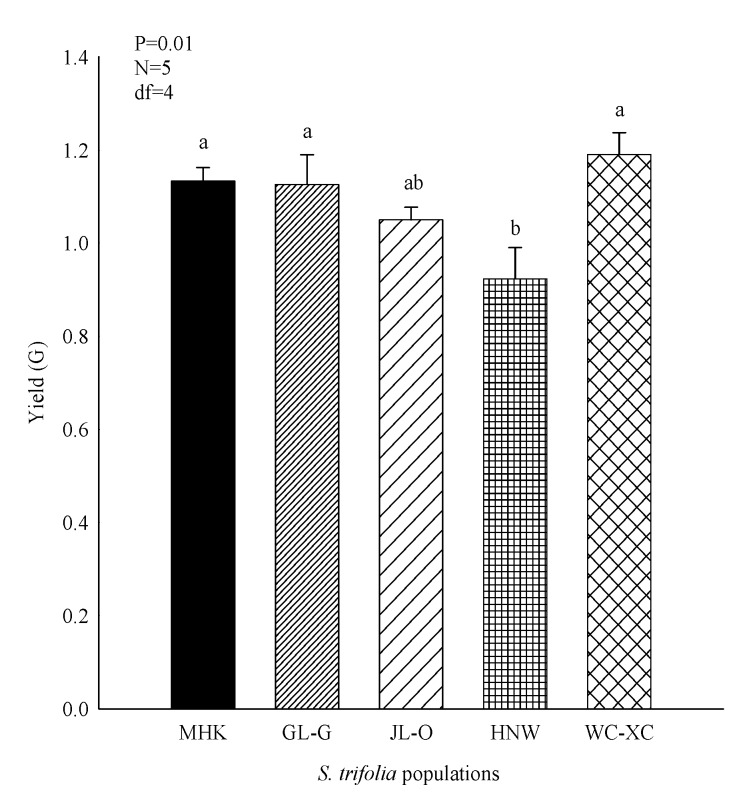
Extract yield (g) obtained from different populations of *S. trifolia*. Values are represented as the mean ± standard error. The same letters on bars indicated that the values are not significantly different according to Tukey’s HSD at the *p* > 0.05 level. N = number of replications; df = degree of freedom.

**Figure 2 molecules-24-03025-f002:**
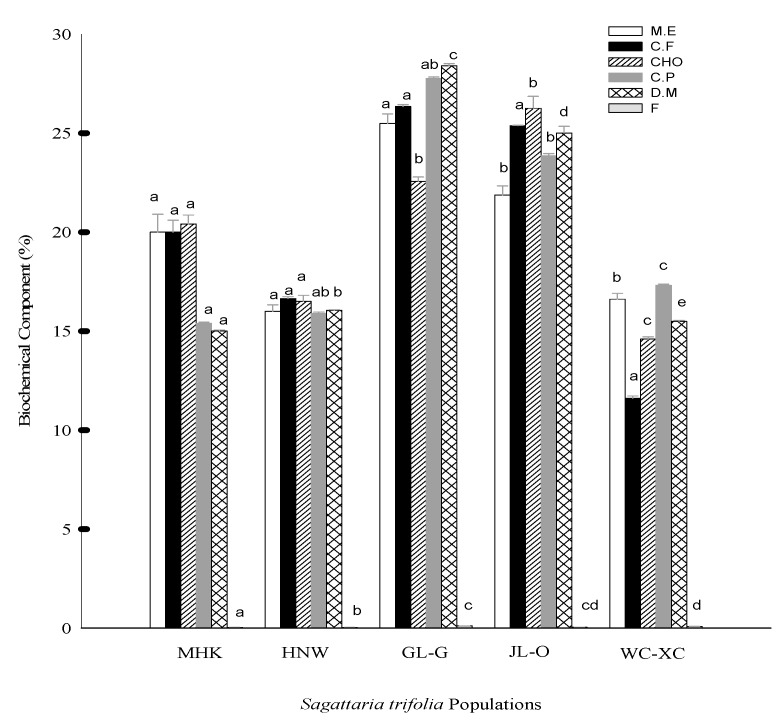
Biochemical composition of *S. trifolia.* Values are represented as the mean ± standard error. The same letters on bars indicated that the values are not significantly different according to the Tukey’s HSD test at *p* ≤ 0.05. M.E (mineral element); C.F (crude fibre), CHO (carbohydrates), C.P (crude protein), D.M (dry matter/ash), F (Fats); MHK (Meihekou), GL-G (Jilin), JL-O (Siping), HNW (Harbin), WC-XC (Wuchang).

**Figure 3 molecules-24-03025-f003:**
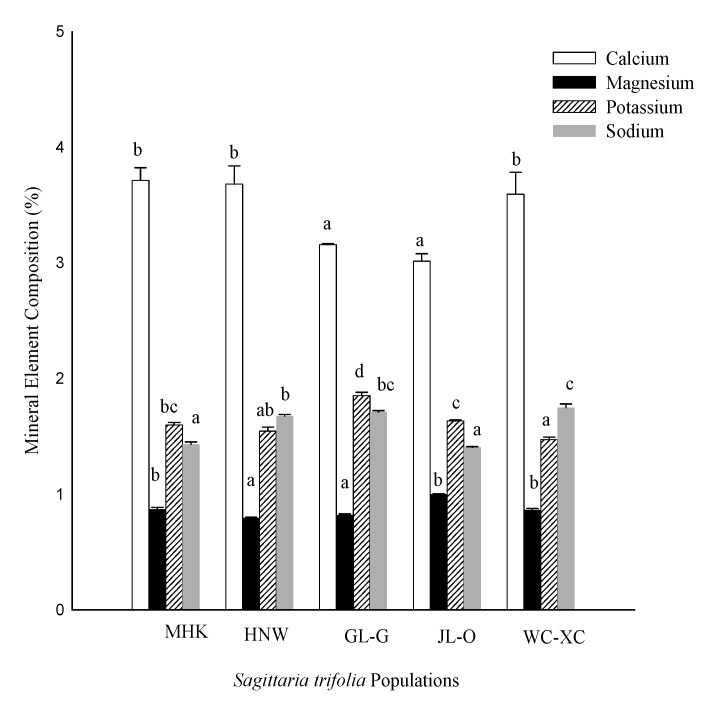
Mineral elements (%) in *S. trifolia* leaves. Values are represented as the mean ± standard error. The same letters within a column indicate that the mean values are not significantly different according to the Tukey’s HSD test at *p* ≤ 0.05. MHK (Meihekou), GL-G (Jilin), JL-O (Siping), HNW (Harbin), WC-XC (Wuchang).

**Figure 4 molecules-24-03025-f004:**
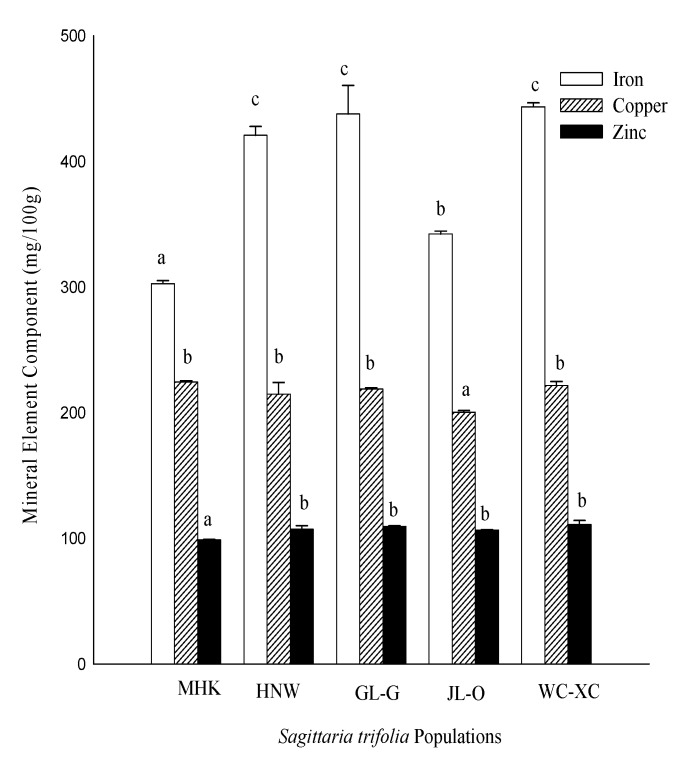
Mineral element (mg/100 g) in *S. trifolia* leaves. Values are represented as the mean ± standard error. The same letters within column indicates that the mean values are not significantly different according to the Tukey’s HSD test at *p* ≤ 0.05. MHK (Meihekou), GL-G (Jilin), JL-O (Siping), HNW (Harbin), WC-XC (Wuchang).

**Figure 5 molecules-24-03025-f005:**
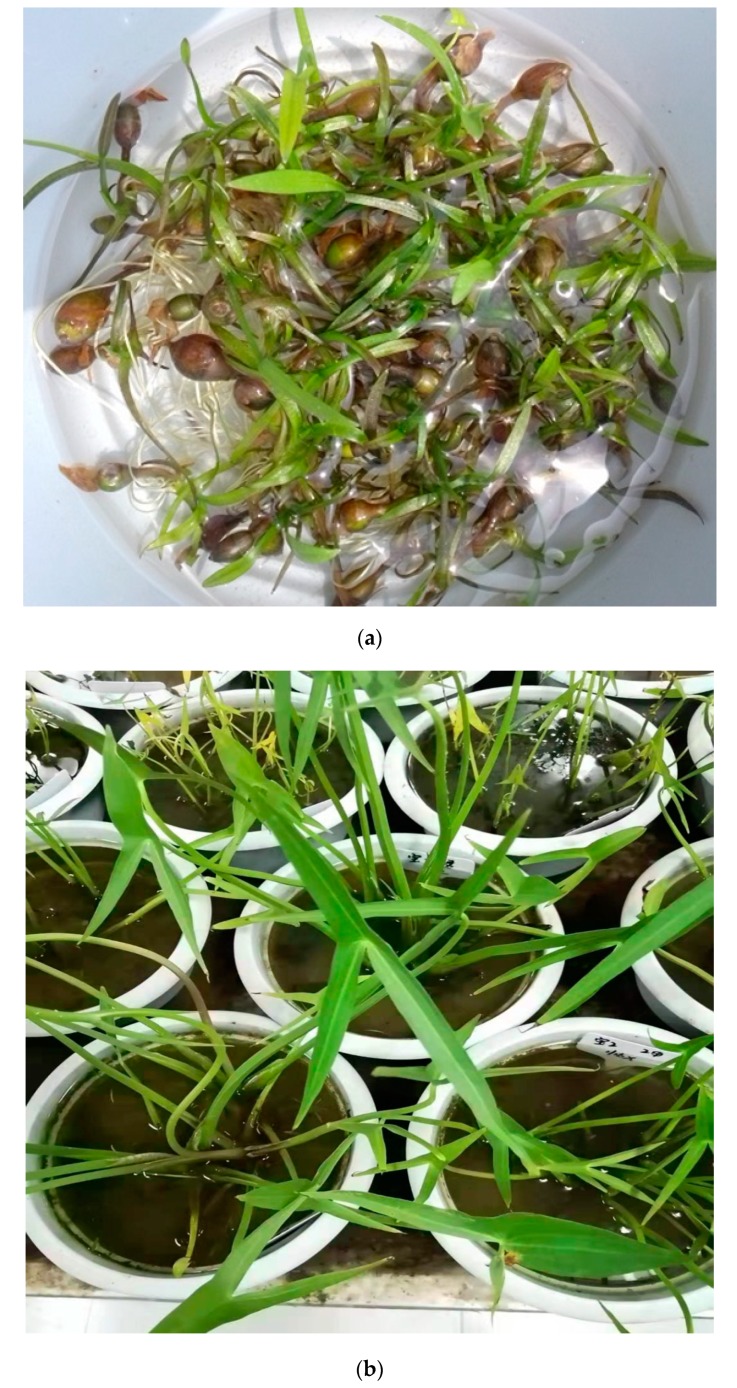
*Sagittaria trifolia*: (**a**) newly sprouted corms; (**b**) plants at 6–7 leaves stage.

**Table 1 molecules-24-03025-t001:** Qualitative phytochemical screening of *S. trifolia* populations.

Treatments (Populations)	Phytochemical Constituents								
	Alkaloids	Tannins	Glycosides	Terpenoids	Steroids	Flavonoids	Flavones	Phenols	Saponins
MHK	−	+	+	−	+	+	+	+	+
GL-G	−	+	+	−	+	+	+	+	+
JL-O	−	+	+	−	+	+	+	+	+
HNW	−	+	+	−	+	+	+	+	+
WC-XC	−	+	+	−	+	+	+	+	+

Where: + presence, − absence; MHK (Meihekow); GL-G (Siping); JL-O (Jilin); HNW (Harbin); WC-XC (Wuchang).

**Table 2 molecules-24-03025-t002:** Total phenolic content (TPC), total flavonoids content (TFC) and DPPH radical scavenging activity of *S. trifolia* populations.

Treatments (Populations)	Total Phenolic Content mg GAE/g	Total Flavonoids Content mg QE/g	DPPH Inhibition (%)
MHK	2.307 ± 0.49 ^a^	12.263 ± 0.49 ^a^	77.347 ± 1.30 ^a^
GL-G	2.050 ± 0.61 ^ab^	10.690 ± 0.36 ^bc^	75.780 ± 0.37 ^a^
JL-O	1.370 ± 0.12 ^c^	9.210 ± 0.19 ^c^	63.670 ± 1.09 ^c^
HNW	1.883 ± 0.82 ^b^	10.390 ± 0.24 ^b^	71.233 ± 1.18 ^b^
WC-XC	2.090 ± 0.57 ^ab^	11.760 ± 0.20 ^a^	67.223 ± 1.74 ^c^
Statistics	*F* = 19.611, *p* = 0.000	*F* = 14.313, *p* = 0.000	*F* = 29.257, *p* = 0.000
summary	S.S = 1.491	S.S = 16.743	S.S = 523.743
	M.S = 4.314	M.S = 130.936	M.S = 0.373
	DF = 4	DF = 4	DF = 4

Values are presented as the mean ± standard error. The same letters within a column indicated that values are not significantly different at (*p* > 0.05) according to Tukey’s HSD; MHK (Meihekow); GL-G (Siping); JL-O (Jilin); HNW (Harbin); WC-XC (Wuchang); GAE (Gallic acid); QE (Quercetin).

**Table 3 molecules-24-03025-t003:** Chemical composition of methanol extract of *S. trifolia* leaves.

Peak. #	Name of Compound	Area %	M.F	M.W
1	Pentanoic acid, 2-[(phenylmethoxy) imino]-, trimethylsilyl ester	7.08	C_15_H_23_NO_3_Si	293.43
2	Ethyl isopropyl ether	1.63	C_5_H_12_O	88.15
3	Cyclohexadien-4-one-1-propiolic acid, methyl ester	0.57	C_10_H_12_O_3_	180.20
4	2,6-Dimethoxytoluene	0.42	C_9_H_12_O_2_	152.19
5	Ethyl Propyl Ketone	0.35	C_6_H_12_O	100.16
6	2-Oxiranecarboxylic acid, 3-(2,2-dimethoxyethyl)-3-methyl-, methyl ester	1.33	C_9_H_16_O_5_	204.22
7	Methyl valerate	0.61	C_6_H_12_O_2_	116.16
8	Furfural/Furaldehyde	2.87	C_5_H_4_O_2_	96.08
9	3-Methoxyphenyl isocyanate	3.09	C_8_H_7_NO_2_	149.14
10	2-Methylbenzoic acid, pentafluorobenzyl ester	1.04	C_15_H_9_F_5_O_2_	316.22
11	Benzoic acid, 4-hydroxy-2-methoxy-3,5,6-trimethyl-, methyl ester	0.42	C_12_H_16_O_4_	224.25
12	Methylhexanoate	0.28	C_7_H_14_O_2_	130.18
13	Dihydromesoanthramine	0.46	C_14_H_13_N	195.26
14	Methyl Thenoate	1.62	C_6_H_6_O_2_S	142.17
15	Octamethylcyclotetrasiloxane	0.22	C_8_H_24_O_4_Si_4_	296.61
16	2-(2-Furyl)-7-methyl-4-quinolinecarboxylic acid	0.62	C_15_H_11_NO_3_	253.25
17	1,3,5,7-Tetraethyl-1-methoxycyclotetrasiloxane	0.46	C_9_H_23_O_5_Si_4_	323.61
18	Hendecane	0.25	C_11_H_24_	156.31
19	Methylbenzoate	0.71	C_8_H_8_O_2_	136.15
20	MethylOctanoate	0.43	C_9_H_18_O_2_	158.24
21	Dodecane	1.91	C_12_H_26_	170.34
22	Hasubanan-9-ol,7,8-didehydro-4,5-epoxy-3,6,6-trimethoxy-17-methyl-,(5.alpha.,9.alpha.,13.beta.,14.beta.)-	0.23	C_20_H_25_NO_5_	359.42
23	Dicetyl	0.23	C_32_H_66_	450.88
24	5-(Hydroxymethyl)-2-(dimethoxymethyl)furan	3.42	C_8_H_12_O_4_	172.18
25	Tetradecane	0.35	C_14_H_30_	198.39
26	Pentadecane	0.34	C_15_H_32_	212.42
27	Phenol, 2,4-bis(1,1-dimethylethyl)	1.47	Unknown	-
28	Cetane	0.46	C_16_H_34_	226.44
29	Glycol Bromohydrin	0.54	C_2_H_5_BrO	124.96
30	Fitone	1.08	C_18_H_36_O	268.48
31	Ethaneperoxoic acid, 1-cyano-1-[2-(2-phenyl-1,3-dioxolan-2-yl)ethyl] pentyl ester	0.39	C_19_H_25_NO_5_	347.41
32	Methylisopalmitate	2.84	C_17_H_34_O_2_	270.45
33	1-Methyl-1-hydroxymethyladamantane	1.55	C_12_H_20_O	180.29
34	Sclarene	3.11	C_20_H_32_	272.47
35	ManoylOxide	5.62	C_20_H_34_O	290.49
36	Phenylacetic acid, 2-methylcyclohex-2-enyl ester	34.30	C_15_H_18_O_2_	230.30
37	Methylstearate	1.26	C_19_H_38_O_2_	298.51
38	3-Octadecyne	7.72	C_18_H_34_	250.47
39	16-Kaurene	5.36	C_20_H_32_	272.47
40	Kaurene-19-ol	3.37	C_20_H_34_O	290.49

M.F (molecular formula); M.W (molecular weight).

**Table 4 molecules-24-03025-t004:** Methods for the determination of biochemical components of *S*. *trifolia.*

Sr. No.	Biochemical Components	Methods	Refs.
1	Dry matter/moisture	1 g of sample was placed in air oven for one day at 105 °C. Moisture content was calculated by before and after weight difference of petri dishes.	[64]
2	Fat	2 g of sample was poured into Soxhlet extraction thimble with addition of 100 mL of petroleum ether in round bottom flask placed at 55 °C for three h until later is siphoned into the receiving flask and was cooled for 10 min. Obtained material was placed in air oven for 60 min at 100 °C and the amount of collected oil (fat) was calculated.	[65]
3	Protein	2 g of sample was mixed with 3 g of digestion mixture (94 g K_2_SO_4_ + 5 g FeSO_4_ + 1 g CuSO_4_) with addition of 12 mL of conc. sulphuric acid (H_2_SO_4_) into the digestion flask. Consequent mixture was heated until it become clear and then on cooling a few drops of 3% boric acid was added for titration followed by few drops of methyl orange and then mixture was completely assorted by continuous hand stirring. Subsequently to this, 10 mL of 0.1N H_2_SO_4_ was taken in burette and titrated until colour changed.	[64]
4	Crude fibre	To calculate crude fibre, briefly, 2 g of sample was dissolved in 200 mL of 1.25% sulphuric acid (H_2_SO_4_) and boiled for 30 min and then filtered. The filtrate was washed with hot water 3–4 times to reduce the acidity. Mixture was re-digested in 200 mL of 1.25% sodium hydroxide (NaOH) and heated, filtered and washed with hot water 3–4 times. After drying the filtrate for 15 min at 100 °C, it was weighed and placed in an electric muffle furnace at 550 °C for 3 h and was re-weighed on cooling for calculating the crude fibre.	[66,67]
5	Carbohydrate	Carbohydrate was calculated by the difference method by subtracting the values of total moisture, mineral, fat, protein and fibre.	[67,68]
6	Mineral element/ash	2 g of sample was placed in electric muffle furnace at 550 °C for 6 h in already weighed porcelain dishes and on cooling reweighed to calculate mineral element/ash.	[66]

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
