# Peer review of "Phytochemical Analysis, Biochemical and Mineral Composition and GC-MS Profiling of Methanolic Extract of Chinese Arrowhead Sagittaria trifolia L. from Northeast China"

_molecules, 2019, doi:10.3390/molecules24173025_

Round 1

Reviewer 1 Report

The manuscript describes substantial results regarding the qualitative and quantitative analysis of Sagittaria trifolia, a native medicinal plant from China. The presented medicinal plant contains forty compounds with antioxidant properties and is a good source of protein, mineral fiber, carbohydrate, and fat content. The Material and method section is clear and the results and discussion section is logically presented. The English language needs to be improved, some sentences are hard to understand, and there are some minor typographical and grammatical errors.

 General observations:

·       The abstract has to be rephrased. It begins with two very long sentences.

·       In the introduction section some sentences need to be rephrased or corrected, ex: line 46 phytochemical -> plural; line 48 which involved -> which are involved; line 52 disease -> plural.

·       Line 77-78 “Its tubers and petioles are cooked in Manipur and Southeast Asia and in Veitnam newly emerged leaves, petioles and tubers are used in the preparation of soup [18].” has to be rephrased. – Petioles and tubers are used twice in the same sentence. Also correct Veitnam -> Vietnam

·       In the results and discussion sections the author uses various references: Marta et al., 2013; Anita et al., 2019; Anita and Pandey, 2017 which do not appear in the references section. Also, format references according to journal guideline!

·       Line 141-142, 165-166 rephrase sentence, line 215 almost -> are almost, line 243 showed -> they showed

·       In the discussion section the sentences should be revised carefully, ex: line 334 different animal -> plural; line 375 these findings are in lined -> in line; line 395 oxygen blinding -> binding; line 398 human -> plural.

·       In the Material and methods section between the number and °C should be a space (line 412, 436, 488-491, etc.)

·       Also a space is missing in line 434 (4mlg-1)

·       Line 515 amount -> the amount

·       The conclusion section should be revised carefully, there are some minor mistakes, ex: line 579 from the present – the is missing, line 580 and can be – and is missing, line 585 human - plural

Overall, the manuscript is of interest and presents detailed work with important results. The grammar and syntax of the manuscript are rather poor and it is very hard to understand the rationale described. The article requires a revision by native English for clear and concise language. Additionally, a correction of the orthography is mandatory.

Author Response

Response to Reviewer 1 Comments

Point 1: The abstract has to be rephrased. It begins with two very long sentences.

Response 1: The abstract has been rephrased and started with short and concise sentences.

Abstract: Sagittaria trifolia is a medicinal foodstuff of China and East Asia belonging to the family Alismataceae. Samples of S. trifolia tubers were collected from Meihekow, Siping, Jilin, Harbin and Wuchang from northeast China. The current study was aimed to evaluate the qualitative and quantitative analysis, antioxidant activity, biochemical analysis and chemical composition of different populations of S. trifolia. By using Folin-ciocalteu, aluminium chloride colourimetric and 1,1-Diphenyl-1-picrylhydrazyl (DPPH), total phenol and flavonoids content and antioxidants activity was analyzed. Furthermore chemical composition, biochemical analysis and mineral substances were also determined. The results showed the presence of flavonoids, phenols, saponins, tannins, glycosides and steroids except for alkaloids and terpenoids by qualitative analysis. Quantitative analysis revealed that highest total phenol, flavonoids content and antioxidant potential identified from Meihekow i.e. 2.307 mg GAE/g, 12.263 mg QE/g and 77.373%, respectively. Gas chromatography-mass spectrometry results showed the presence of fourty chemical compounds corresponding to 99.44% of total extract that might be responsible for antioxidant properties. Mineral and biochemical analysis revealed the presence of Calcium, Magnesium, Potassium, Sodium, Iron, Copper, Zinc and carbohydrate, protein, fiber and fat contents respectively. Interestingly, all S. trifolia populations collected from different locations possess similar composition. The dietary values, phytoconstituents, antioxidant activities and nutritional and curative chemical compounds of S. trifolia are beneficial for the nutritherapy of human beings

Point 2: In the introduction section some sentences need to be rephrased or corrected, ex: line 46 phytochemical -> plural; line 48 which involved -> which are involved; line 52 disease -> plural.

Response 2: In the introduction part sentences and words has been rephrased and corrected as suggested such as; line 45; Natural phytochemical -> Phytochemicals

which  involved-> which are involved

line 52; disease -> diseases

Point 3: Line 77-78 “Its tubers and petioles are cooked in Manipur and Southeast Asia and in Veitnam newly emerged leaves, petioles and tubers are used in the preparation of soup [18].” has to be rephrased. – Petioles and tubers are used twice in the same sentence. Also correct Veitnam -> Vietnam.

Response 3: Line 77-78 rephrased; Cooked tubers and petioles are used as edible purposes in Manipur and Southeast Asia while, in Vietnam these parts of S. trifolia are used in the preparation of soup. Veitnam-> spell checked and corrected as Vietnam.

Point 4: In the results and discussion sections the author uses various references: Marta et al., 2013; Anita et al., 2019; Anita and Pandey, 2017 which do not appear in the references section. Also, format references according to journal guideline!

Response 4: In the results and discussion section Marta et al., 2013 has been removed from the text. While, Anita et al., 2019; Anita and Pandey, 2017 has been cited as Rao et al., 2019 [22] and Rao and Panday, 2017 [24] according to journal style.

Point 5: Line 141-142, 165-166 rephrase sentence, line 215 almost -> are almost, line 243 showed -> they showed.

Response 5: Sentences in line 141-142 and 165-166 has been rephrased. Biochemical analysis section has been converted into table form and data has been concise too.

Point 6: In the discussion section the sentences should be revised carefully, ex: line 334 different animal -> plural; line 375 these findings are in lined -> in line; line 395 oxygen blinding -> binding; line 398 human -> plural.

Response 6: corrected like line 334 (272) naturally in different animals and in oils; Lined changed to line; 398 (324) oxygen blinding changed to oxygen binding. 398 (333) increasing demands of human beings globally.

Point 7: In the Material and methods section between the number and °C should be a space (line 412, 436, 488-491, etc.).

Response 7: In the whole section of M & M a space has been inserted between number and °C followed as suggested.

Point 8: Also a space is missing in line 434 (4mlg-1).

Response 8: corrected 434 -> (370) “(4 mlg-1 of sample)”

Point 9: Line 515 amount -> the amount.

Response 9: corrected as suggested and presented in Table 4 such as “at 100 °C and the amount of collected oil (fat) was calculated

Point 10: The conclusion section should be revised carefully, there are some minor mistakes, ex: line 579 from the present – the is missing, line 580 and can be – and is missing, line 585 human – plural.

Response 10: in conclusion section some revisions like 579 –> (497) from the present;

Line no.  580 -> (504) corrected as and could be engaged; line 585 (511) revised.

Conclusion: Results from the present study suggested that S. trifolia contain high proportion of phytoconstituents, total phenol and flavonoid content thereby conferring to the free radicals scavenging activities and can be a potent source of natural antioxidant. Chemical composition revealed the presence of fourty compounds responsible for antioxidant potential of S. trifolia. Furthermore, it’s leaves are considered to be a good source of carbohydrates, protein, fiber, fat content and minerals typically, Calcium, Magnesium, Potassium, Sodium, Iron, Copper and Zinc in appreciable amount. Hence, S. trifolia could be a contributor towards meeting nutritional requirements of humans if used in adequate amount and could be considered as positive source of defense for protection against various diseases. However further studies are desired to explore its dietary potential for the sake of commercial purpose and toxicity studies.

Reviewer 2 Report

Reviewer’s comments on Molecular 561278

In this work, the authors analysed the antioxidant activity, biochemical analysis and chemical composition of Sagittaria trifolia. The mineral substances were also determined. By using Folin-ciocalteu, aluminium chloride colourimetric method and 1,1-Diphenyl-1-picrylhydrazyl (DPPH), total phenol and flavonoids content and antioxidants activity was analyzed. Alkaloids and terpenoids were not detected, the existence of flavonoids, phenols, saponins, tannins, glycosides and steroids were detected by qualitative analysis. Maximum total phenol, flavonoids content and antioxidant potential were reported. GC-MS results explored the existence of fourty chemical compounds corresponding to 99.44% of total extract responsible for antioxidant properties. Mineral and biochemical analysis revealed the presence of calcium, magnesium, potassium, sodium, ferrous, copper, zinc and, carbohydrate, protein, fiber, mineral and fat contents, respectively. Interestingly, all Sagittaria trifolia collected from difference resource have almost the same composition. The dietary values, phytoconstituents, antioxidant activities and nutritional chemical compounds are benefit for the nutritherapy of human beings. Systematical work has been done in the current paper; it provides chemical component information for food quality evaluation. There are several places need to be addressed:

1.      In table 3, there are some mistakes in the name of the compounds, such as “Benzene, 1-isocyanato-3-methoxy-”; “9-Anthracenamine, 9,10-dihydro-”; “Cyclotetrasiloxane, octamethyl- ”,the authors please check the table carefully.

2.      The fluoride containing molecules 2-Methylbenzoic acid, pentafluorobenzyl ester is not stable in the solution conditions, it will be hydrolysed to the acid and pentaflurobenzyl alcohol, the authors please rationalize the formation of this compounds, or cite more references.

3.      The format of references needs to be organized. “[21] Wu Leung, W.T.; Rauanheimo Butrum, R.; Huang Chang, F.; Narayana Rao, M.; Polacchi, W. Food composition table for use in East Asia; Food & Agriculture Org.: 1972.”; “[29] Yoshikawa, M.; YAMAGUCHI, S.; Murakami, T.; MATSUDA, H.; YAMAHARA, J.; MURAKAMI, N. Absolute stereostructures of trifoliones A, B, C, and D, new  biologically active diterpenes from the tuber of..” etc. the authors please carefully check the references.

Author Response

response to reviewer 2 is attached as separate file

Reviewer 3 Report

Paper review: Phytochemical analysis, biochemical and mineral composition and GC-MS profiling of methanolic extract of Chinese arrowhead Sagittaria trifolia 3 from northeast China

Minor and major remarks:

please revise abstract - it should be A PILL of new developments as well as new data/results (please do not describe method which were used for determination)

please create in points aims of the research at the end of introduction section with (several bullet points)

please indicate what is the novel in these research based on actual state of knowledge (why this research is important and what is the address of this research - only China or whole world?)

authors should decide what will be better - text or figure - please do not duplicate both, do not use both forms like e.g. in 3.1 (1 sentence with values is the same like on fig. 2) - This remark is connected with whole manuscript content

please remove fig. 3 - table 3 is enough

deep revision is necessary for Discussion section (lack in citations)

Materials and methods:

- remove the equations and add the citation for calculations

- please carefully check the reagents formulas e.g. - Al2Cl3 ? and correct as well as symbols (Fe - IRON not ferrous)

- if authors cite the specific method please do not describe it as a copy of original method

In general rebuild M&M section for wet methods and analytical methods, all of these methods are not new - please do not describe it - only cite (I suggest to prepare table).

In General, manuscript need deep revision as well as language correction (with specialist chemistry language). I suggest to add also information/data on toxic metals in S. trifolia and toxic organic compounds to have a knowledge about potential usage in food industry. However, quality of the manuscript is low, validation of presented method has not been presented as well, conclusions are not supported with the results and finally I cannot find any documented research novelty in this manuscript.

Author Response

Comments to reviewer 3 has been attached as separate file

Reviewer 4 Report

The authors studied the qualitative and quantitative analysis, antioxidant activity, biochemical analysis and chemical composition of different populations of S. trifolia. The work was well presented. Here are some of my concerns. 

Since they collected the samples and cultured them in a greenhouse, I'm wondering if the authors considered the local environmental influences on the S. trifolia growth. In my opinion, the greenhouse changed the original growth environment of S. trifolia. 

The format of the manuscript was unordered.

Why was Figure 1 at the end of the manuscript?

Some writing used Fig. some used Figure. 

Suggest the authors take time to regulate their manuscript. 

Very confusing writing "Total phenol and flavonoids content and antioxidants activity was assessed by Folin-ciocalteu, aluminium chloride colourimetric method and 1,1-Diphenyl-1-picrylhydrazyl (DPPH) respectively while, chemical composition, biochemical and mineral analysis were determined by Gas chromatography-mass spectrometry (GC-MS), AOAC standards methods and NOVA 400 atomic absorption spectrometer respectively."

Suggest the authors rewrite the introduction part. 

Author Response

Comments to reviewer 4 attached as separate file 
